# Toughening and Healing of CFRPs by Diels–Alder-Based Nano-Modified Resin through Melt Electro-Writing Process Technique

**DOI:** 10.3390/ijms23073663

**Published:** 2022-03-27

**Authors:** Athanasios Kotrotsos, George Michailidis, Anna Geitona, Filippos Tourlomousis, Vassilis Kostopoulos

**Affiliations:** 1Department of Mechanical Engineering and Aeronautics, University of Patras, GR-26504 Patras, Greece; akotrotso@mech.upatras.gr (A.K.); michasg397@gmail.com (G.M.); annageitona@gmail.com (A.G.); 2Biological Lattice Industries, National Center for Scientific Research ‘Demokritos’, GR-15341 Athens, Greece; filippos.tourlo@cba.mit.edu; 3Foundation of Research and Technology, Institute of Chemical Engineering Sciences (FORTH/ICE-HT), Stadiou Str., GR-26504 Patras, Greece

**Keywords:** self-healing, electrospinning, melt electro-writing, Diels–Alder reaction, bis-maleimides, mode I testing, graphene nanoplatelets, healing efficiency, optical microscopy, c-scan

## Abstract

In the current study, a novel approach in terms of the incorporation of self-healing agent (SHA) into unidirectional (UD) carbon fiber reinforced plastics (CFRPs) has been demonstrated. More precisely, Diels–Alder (DA) mechanism-based resin (Bis-maleimide type) containing or not four layered graphene nanoplatelets (GNPs) at the amount of 1 wt% was integrated locally in the mid-thickness area of CFRPs by melt electro-writing process (MEP). Based on that, CFRPs containing or not SHA were fabricated and further tested under Mode I interlaminar fracture toughness experiments. According to experimental results, modified CFRPs exhibited a considerable enhancement in the interlaminar fracture toughness properties (peak load (P_max_) and fracture toughness energy I (G_IC_) values). After Mode I interlaminar fracture toughness testing, the damaged samples followed the healing process and then were tested again under identical experimental conditions. The repeating of the tests revealed moderate healing efficiency (H.E.) since part of the interlaminar fracture toughness properties were restored. Furthermore, three-point bending (3PB) experiments were conducted, with the aim of assessing the effect of the incorporated SHA on the in-plane mechanical properties of the final CFRPs. Finally, optical microscopy (OM) examinations were performed to investigate the activated/involved damage mechanisms.

## 1. Introduction

During the last decades, carbon fiber reinforced composites (CFRPs) have become the preferable materials for many advanced structures because they offer high specific strength and stiffness, low weight and high fatigue and corrosion resistance. However, laminated composites, such as CFRPs during service life, are susceptible to delamination, mainly due to the absence of reinforcement in the out of plane direction that leads to separation of plies (particularly under compressive loading conditions) [1]. Other geometrical irregularities in composite structures that promote the initiation of delamination could be the following: holes, cut-outs, stiffener terminations, sandwich panels, ply drops, free edges, and bolded and bonded joints [2]. In addition to these, the contamination of fibers during manufacturing of the composite, the shrinkage of the matrix, and the insufficient wetting of the reinforcement itself are some other reasons for delamination promotion. Based on these, Mode I interlaminar fracture toughness characterization [3,4] is of great importance for structural design of composites structures.

Lately, various researchers have aimed to extend the effective life-span of the composites. Self-healing (SH) is an emerging technology for this type of materials, but is not yet commercially applied [5]. Existing SH mechanisms are divided into two main categories: (a) extrinsic type (vascular or capsule based mechanisms) [6,7,8,9] and (b) intrinsic type (reversible polymers strategy) [10,11,12,13,14,15,16]. In the first category, self-healing agent (SHA) release occurs during the crack propagation process within a composite structure. After release, the liquid SHA (Part A) will be in contact with a solid or a liquid catalyst (Part B) within the polymer matrix. Then, the polymerization process commences, which further leads to the solidification of the SHA itself [17,18]. The second category comprises thermal reversible polymers that allow the healing to be unlimited if compared to extrinsic type while the healing process takes place after heating of the materials beyond its glass transition (T_g_) or melting point (T_m_) temperature. Furthermore, hollow containers usually act as stress concentration points [6] as a result the degradation of the in-plane mechanical properties of the materials.

Diels–Alder (DA)-based materials present resin-type behavior while they are mendable and they also present healing functionality through heating upon the T_m_ temperature [19,20]. This type of materials is based on special covalent bonding interactions that allow polymer chains to flow upon heating and have the potential to heal internal cracks within the composites’ matrix [4,10,11,12,16]. Recently, Smojver et al. [21] presented a micromechanical constitutive model of an intrinsically mendable CFRP. The utilized specimens consist of a bis-maleimide (BMI) tetrafuran polymer (trade code: 2MEP4F), and unidirectional carbon (UD) fibers. The model was implemented into Abaqus and validated using experimental results of three-point bending (3PB) tests. On the other hand, common thermoplastics have the potential to melt due to the simple covalent bonding interactions and can be easily selected as potential SHA for the fabrication of healable composites despite the poor mechanical properties [13,14,22]. Another promising category of healable polymers is also those who are based on supramolecular chemistries (non-covalent interactions) [3,15,23,24]. Supramolecular polymers usually present high healing efficiencies (H.E.) when incorporated into composites, but cause degradation of the in-plane mechanical properties [24]. The degradation effect occurs due to the weak nature of supramolecular polymers’ bonds (i.e., hydrogen bonds, etc.).

Since 2010 [25], the use of a solution electrospinning process technique for the fabrication of healable structures (i.e., polymers and polymeric composite materials, coatings and batteries) is a common practice. On the other hand, the use of a melt electro-writing process (MEP) technique for SH purposes is still absent. Until now, MEP was mainly utilized for biomedical applications and especially for the fabrication of three-dimensional (3D) scaffolds [26]. MEP is a processing technique for producing fibrous structures from polymer melts [27]. In this particular case of the electrospinning technique, the collection of the fiber can be very focused. Therefore, combined with moving collectors, MEP is a way to perform 3D printing with ultrathin fibers. In our case, ultrathin UD fibers of DA resin were locally and directly printed on the surface of a UD carbon fiber pre-preg ply. In [28], the authors utilized the 3D printing method to fabricate samples with the same DA material in order to investigate the mechanical properties and the healing behavior. According to experimental results, the printing orientations did not affect samples’ mechanical performance as well as their SH capability because of the isotropic nature of the prints.

The current work deals with a novel way of SHA incorporation into a UD composite structure. More precisely, high performance CFRPs were locally modified in their mid-thickness area by DA-based SHA nano-modified or not with GNPs through MEP technique. After the manufacturing process, reference and modified samples were tested under Mode I loading experimental conditions and further evaluated. Experimental results revealed that the integration to the SHA led to a significant increase in the Mode I fracture toughness properties of the fabricated CFRPs with the nano-modified ones (with GNPs) to exhibit the best toughening behavior. After fracture, both modified CFRPs followed the healing procedure to activate the healing effect and then tested again under the same experimental conditions. After retesting, modified samples exhibited acceptable H.E., taking into consideration previous publications [4,11,16] conducted by the authors. Finally, 3PB tests revealed no significant degradation effects due to SHA incorporation while optical microscopy (OM) examinations of the fractured surfaces and cross-sectional area of the samples led to qualitative results regarding the activated/involved failure mechanisms.

## 2. Results and Discussion

### 2.1. Test Outline Program

Initially, the reactants of the fabricated SHA (i) the trifuran (TF) and (ii) the BMI oligomers (BMI-1700) were put together in stoichiometric analogy and left into an oven to react and for the cross-linking effect to be further achieved. TF synthesis procedure is provided in a previous publication by the authors [12] while BMI-1700 oligomers were commercially available. In the case of the GNP-modified SHA, 1 wt% of the nano-filler was homogeneously dissolved into the SHA prior to the cross-linking reaction. After SHA synthesis, the steps followed for the current investigation are the following: (a) integration of the SHA (nano-modified or not) by MEP on the pre-pregs’ surface locally and only at areas of interest; (b) manufacturing of all CFRP plates and further curing of them by using an autoclave; (c) quality control of all CFRPs; (d) conduction of Mode I interlaminar fracture toughness and 3PB tests; and (e) activation of the SH procedure and repeating of the tests by applying identical experimental conditions. Figure 1 schematically provides the testing routine conducted for the needs of the current investigation.

### 2.2. Composites’ Manufacturing and Quality Issues

Taking into consideration the c-scan inspections conducted for all material sets (reference and both modified CFRPs) manufactured, it was shown that all materials presented an absence of defects (i.e., porosity, delamination, etc.) due to manufacturing. C-scan plots for all CFRP plates are provided in Section 3.4. On the other hand, Figure 2 illustrates optical microscopy (OM) images of cross-sectional areas of a reference and a representative modified sample. CFRPs containing SHA (nano-modified or not) provided the same OM images for the cross-section areas as anticipated. As is clearly illustrated, modified samples (Figure 2b) presented local darkening of the matrix into the mid-thickness area due to SHA presence, unlike reference ones (Figure 2a). The mid-thickness area represents the area in which the crack will be propagated during Mode I testing and where the healing effect will be further taken advantage of.

### 2.3. Mode I Testing

In the current section, the quasi-static Mode I tests which were performed for all material sets (reference and modified CFRPs) are fully provided and further discussed. The tests were conducted according to the given specifications of Section 3.5. The incorporation of the SHA (nano-modified or not) into composites architecture is expected to have an impact on the Mode I interlaminar fracture toughness properties of the double cantilever beam (DCB) samples. The main purpose of the present experimental work at this stage was (a) to identify the most effective electrospun SHA which provided the best toughening behavior and (b) to evaluate the effect of GNPs on the final Mode I fracture toughness properties of the CFRP.

In Figure 3a, representative load (P) versus displacement (d) curves are provided for all material sets exposed to Mode I loading conditions. For all CFRPs, the general trend does not present any difference as the applied load was linearly increased, followed by a deviation from the linearity and ended by a load drop after the crack propagation onset. Nevertheless, it is of note that the two modified CFRPs can bear a significantly higher load during Mode I testing. These materials, presented an unstable “stick-slip” crack growth, whereas, in the case of the reference CFRP, a relatively smoother behavior is observed in the P vs. d curves. According to experimental results, it was shown that the two modified CFRPs presented noticeably enhanced fracture toughness properties when compared to the reference one. Thus, the SHA incorporation has the potential to toughen the final CFRP. An analogous behavior was also shown in [4,11,16] as DA-based SHAs usually have the ability to increase the toughening performance of composites. Finally, it is also clear that the apparent stiffness of the composites does not present any noticeable difference.

Figure 3b compares the Mode I interlaminar fracture toughness properties (P_max_ and G_IC_ values) for all CFRPs. It shows that the CFRP samples containing the nano-modified SHA (with 1 wt% GNPs) exhibited the best toughening profile. More precisely, CFRPs containing pure DA-based material (SHA without GNPs) presented 96.7% (from 96.7 ± 4.09 N to 190.2 ± 21.54 N) and 145.7% (from 0.33 ± 0.06 kJ m^−2^ to 0.81 ± 0.07 kJ m^−2^) higher P_max_ and G_IC_ values, respectively. On the other hand, samples containing BMI SHA and 1 wt% GNPs exhibited an increase of 129.3% (from 96.7 ± 4.0 N to 221.7 ± 23.25 N) for P_max_ value and 284.1% (from 0.33 ± 0.06 kJ m^−2^ to 1.26 ±0.41 kJ m^−2^) increase for G_IC_ value.

In Figure 4a, a comparison between the crack opening resistance curves (R-curves) for all material sets is given. As it is clearly shown, the capacity of DA SHA itself and GNPs at the amount of 1 wt% to significantly improve the Mode I fracture toughness properties is noticeable. Reference CFRPs presented a limited damage process zone (DPZ), lower than 5 mm, while they reached a plateau just after a crack propagation length (CPL) of 30 mm. On the contrary, the two modified CFRPs presented an enhanced DPZ. The BMI-modified CFRPs exhibited at least 20 mm larger DPZ against reference CFRP, while it reached a plateau just after a CPL of 55–60 mm. For the BMI- and GNP-modified samples the fracture mode slightly differed as these samples exhibited higher DPZ against the two other CFRPs (reference and BMI-modified CFRPs) without reaching a “strong” plateau. More precisely, BMI- and GNP-modified CFRPs exhibited at least 40 mm larger DPZ compared to the reference ones, while the curve reached a plateau just after a CPL of 70 mm. Finally, all CFRPs promoted relatively stable growth of the CPL, as Figure 4a implies. In Figure 4b, a photograph during the Mode I test of a modified CFRP sample is illustrated.

The enhancement of the Mode I fracture toughness properties for both modified CFRPs is strongly attributed to the bridging effect between the two crack flanks (top and bottom) due to SHA presence into the mid-thickness area of the final CFRP (see Figure 5b). The main process behind this is the suppression of the crack tip opening stresses, which in its turn resulted in a reduction in the crack opening displacement at a given applied load against the reference CFRP. In addition, the incorporation of a ductile phase into the mid-thickness area in fibrous form as a SHA material considerably enhanced the phenomena that are related to plastic deformation mechanisms. Thus, the combination of the bridging mechanism together with the plastic deformation phenomena led to the significant improvement of the mechanical properties under Mode I testing, as more energy is required. According to the relevant literature [4,16], DA mechanism-based SHAs, when incorporated into mid-thickness area, have the ability to enhance the fracture toughness I properties of the final CFRP. In addition to this, the strong presence of GNPs (at the amount of 1 wt%) into the SHA structure provided an additional enhancement of the fracture toughness properties, as the result was that samples containing GNPs into SHA presented the best toughening profile. The incorporation of these nano-fillers has shown to improve the interlaminar fracture toughness properties of the composites. In a recent paper by the authors [16], it was shown that modified CFRPs containing the same nano-modified SHA were able to significantly increase the fracture toughness properties of the CFRPs. In this case, the modification of CFRPs was performed by the solution electrospinning process technique. Apart from GNPs, carbon nano-tubes (CNTs) also have the potential to toughen CFRPs. In [13], Kostopoulos et al. achieved a considerable increase in both Mode I and Mode II interlaminar fracture toughness properties of CFRPs by modifying them with CNTs reinforced nylon micro-particles. On the contrary, reference CFRPs showed a clear brittle behavior, as shown in Figure 5a.

### 2.4. Repair of the Delaminated CFRPs Via the Healing Treatment

After first fracture, the delaminated samples passed through the healing process, which consisted of uniform heating and compression, in order to be repaired. The applied healing profile is provided in paragraph 3.6. Reference CFRPs, having no healing functionality, were not subjected to the healing process. According to a recent publication by the authors [16], it was proven that when a reference CFRP is subjected to the precise healing cycle, no healing occurs as anticipated based on c-scan inspections.

Figure 6 provides representative P-d curves for all material sets exposed to Mode I loading conditions before and after the healing process (Figure 6a for BMI-modified samples and Figure 6c for BMI- and GNP-modified samples). In addition, bar chart diagrams of Figure 6 contain the synopsis of the Mode I fracture toughness properties (P_max_ and G_IC_ values) for both modified samples prior to and after the activation of the healing cycle (Figure 6b for BMI-modified samples and Figure 6d for BMI- and GNP-modified samples) together with H.E. values. For all material sets, the general behavior (prior to and after the healing process) does not show any difference, as the applied load was initially linearly increased. Afterwards, the load deviated from the linearity and ended with a drop after the crack propagation onset. Nevertheless, it is of note that the healed CFRPs are able to bear significantly lower load during testing.

After first fracture and following the healing process, the only material that can act and heal the delaminated area of the CFRP is the DA material which was placed into the mid-plane area through MEP. This behavior occurs due to SHA ability to melt and reform connections during the healing process through retro DA reaction mechanism. Thus, the results from the tests followed the first damage are solely controlled by the structural properties of the additive as well as its compatibility with the constituent materials. The P_max_ and G_IC_ values for both sets of CFRPs from the two tests and the corresponding H.E. achieved after testing are summarized in Figure 6b,d for BMI-modified and BMI- and GNP-modified CFRPs, respectively. Based on experimental results, BMI-modified CFRPs can recover 30% and 3% of its initial P_max_ and G_IC_ values. Interestingly and even better, BMI- and GNP-modified samples can recover 62% and 45% of its initial P_max_ and G_IC_ values.

It is also of note that the apparent stiffness of the two mendable CFRPs decreases with the application of the healing cycle. The decrease in the stiffness could be attributed to the extended loads and displacements which progressively damaged the outer plies of the specimens that are highly loaded. A similar observation is also reported in analogous study [13]. Finally, another important observation was that nano-modified samples exhibited higher H.E. values if compared to them without nano-fillers. This could probably be attributed to the presence of the thermally conductive GNPs which promote better melting in local scale within the composite structure. An analogous behavior was also observed in a relative study by the authors [13], in which the utilized nano-filler was multi-walled CNTs.

The typical R-curves under Mode I loading conditions prior to and after the application of the healing cycle are depicted in Figure 7a,b for BMI-modified and BMI- and GNP-modified samples, respectively. The general trend for both material sets is that after the activation of the healing process the G_I_ values significantly decrease (especially for BMI-modified CFRPs which exhibited lower H.E. values) and the plateau values of the G_I_ are reached earlier for BMI-modified CFRPs (at 40 mm instead of 60 mm) than the corresponding plateau values of the initial CFRPs (pristine BMI-modified samples). For BMI- and GNP-modified samples, after healing process, the plateau was reached almost at the same point (close to 70 mm).

### 2.5. Effect of Self-Healing Agent and Nanofillers on In-Plane Mechanical Performance of CFRP Structure

The incorporation of both types of SHA (nano-modified or not) into mid-thickness area of the CFRPs is expected to affect the flexural properties (in-plane) of the final composites. Based on that, 3PB tests were conducted, taking into consideration the given guidelines of paragraph 3.7. The tests were conducted just after the manufacturing process and prior the healing process.

Bar chart of Figure 8 provides and compares the flexural properties between reference and modified CFRPs in order that the effect of the SHA incorporation (in nano-modified form or not) can be identified. More precisely, Figure 8 provides the flexural modulus (E_Flex_) and flexural strength (σ_max_) values for all CFRPs. According to experimental results, it was shown that, by incorporating SHA, the E_Flex_ was not significantly affected, as it was slightly decreased by 6.9% (from 128.26 ± 5.4 GPa to 119.51 ± 0.96 GPa) and 4.3% (from 128.26 ± 5.4 GPa to 122.69 ± 0.91 GPa) for BMI and BMI- and GNP-modified CFRPs, respectively. On the other hand, the σ_max_ value was not affected, as a slight increase of almost 2% was observed for both modified CFRPs. The slight increase in the σ_max_ value is in the experimental error margins and is considered to be out of insignificance. The slight decrease in E_Flex_ values was anticipated, as the incorporated SHA (having lower mechanical properties) during the curing process has locally replaced part of the host epoxy matrix. Similar behavior was also shown in a recent study [16].

## 3. Materials and Methods

### 3.1. Materials

The utilized materials for the needs of the current investigation (i.e., the pre-preg tape type, the maleimide oligomers, the reactants for the TF synthesis and the GNPs) as well as their specifications are fully provided in a recent publication by the authors [16].

### 3.2. Preparation of the SHAs

In this work, the cross-linked BMI network (BMI-1700), as well as the doped one containing GNPs at the amount of 1 wt%, played the SHA role into mid-thickness of CFRPs. For the pure SHA preparation, stoichiometric amounts of BMI-1700 oligomer and TF compound were mixed and placed into a silicon mold. The TF synthesis procedure is described in detail in [12] and is based on a procedure conducted by Ling et al. [29]. After TF synthesis, nuclear magnetic resonance (NMR) and Fourier transform infrared spectroscopy (FTIR) tests were conducted in order that the chemical structure of the resulting TF material could be identified. According to these tests, it was shown that NMR and FTIR results were proven to be the same as those in [29], and more information can be found there.

After the mixing of the reactants, the system was transferred into an oven. The utilized heating profile of the oven was the following: the mixture was exposed to 80 °C for 3 h and then was left to cool down to room temperature (RT) conditions. During heating of the mixture, the cross-linking reaction (DA reaction) between BMI-1700 and TF took place. In [4], the same SHA type was utilized to impregnate carbon fiber fabrics in order that self-healable pre-pregs could be fabricated and further incorporated into a CFRP structure. For the preparation of the GNP-modified SHA network, initially the GNPs were dispersed into tetrahydrofuran (THF) solvent and then sonicated by using the Bandelin (Berlin, Germany) electronic sonicator apparatus for 3 h, at a selected frequency of 35 kHz to break down potential agglomerates. Then, the BMI-1700 oligomer and the TF compound were dissolved into the GNPs suspension and were magnetically stirred under heating at 0.8 of THF boiling point (b.p.^(THF)^ ~40 °C) to achieve good homogeneity, while, in parallel, the solvent was evaporated. After stirring, the nano-modified mixture passed through the same heating procedure as in the case of pure SHA fabrication in order that the cross-linking effect could be achieved. After cross-linking, all SHA types (pure and nano-modified) were cut into small pieces in order to be inserted into the metallic hopper of the MEP set-up and to further draw fibers onto pre-pregs’ surface.

### 3.3. Melt Electro-Writing Process and Preparation of the Modified Pre-Preg Ply

The incorporation of the SHA into CFRPs was performed by using the MEP technique. MEP was directly applied onto pre-pregs’ surface, and more precisely at targeted areas (locally). The reason why the incorporation of the SHA into composites was applied locally was dual: (a) for material economy and (b) to eliminate potential knock-down effects. The MEP set-up consists of a heated metallic hopper (melt container), a high voltage power supply, a capillary (needle) with an inner diameter of 0.5 mm and a grounded collector. The temperature of the metallic hopper was controlled by a temperature control system. Each pre-preg ply was placed on collector’s surface in order for the direct deposition of the SHA fibers on it to be achieved. SHA was transferred into the metallic hopper after cutting of it into small pieces as earlier mentioned. The applied temperature was raised to 110 °C until the polymer droplet (melted polymer) was formed. At this stage, the Retro DA reaction (reverse DA-reaction) between TF compound and BMI-1700 occurs, resulting into a viscous SHA drop at the needle tip. After droplet formation, an electric field was generated between the tip of the needle and the collector by applying high voltage to the system having as a result of a jet formation that elongates through the electric field and finally deposits directly on the pre-preg’s surface (pre-preg surface represents collector at this stage). During MEP, no additional pressure was applied to the molten polymer to pass through the needle. The deposition took place at ambient conditions. The distance between the needle and the collector was 3 mm and the applied voltage was kept at 4 kV during polymer fibers’ deposition. After the MEP procedure, the final areal weight of the fibrous structure (modified area) was calculated to be approximately 48.8 g/m^2^ (or 0.0488 g/cm^2^) and the capacity of MEP “writing” for the specific polymeric network was 1.22 g/h. The geometrical features of the fibrous structure written on pre-pregs’ surface is illustrated in the OM image of Figure 9d. Based on this image, UD fibers of the DA material have been drawn with an average diameter of 150 ± 10 μm, while the distance between the fibers was found to be close to 250 ± 10 μm.

### 3.4. Composites Manufacturing, Quality Control and Optical Microscopy Examinations

For the composites’ manufacturing, UD carbon pre-pregs were utilized while the curing process was performed by using autoclave technologies. Three types of laminated plates with 22 UD pre-pregs were fabricated: the reference and two modified laminates containing the electrospun SHA (with and without 1 wt% GNPs) into their mid-thickness area. During MEP, the deposited SHA onto pre-pregs’ surface created a “fibrous interleaf” with the aim of providing healing functionality to the final CFRP. Figure 9a,b illustrate the design of the mid-thickness area of the fabricated plates and a representative surface-modified pre-preg tape after MEP, respectively.

Three plates were manufactured with dimensions 270 mm × 150 mm × 3 mm that led to 6 DCB samples for each category, with final dimensions of 250 mm × 25 mm × 3 mm. According to the AITM 1.0005 standard of Airbus [30], the shape of the samples is illustrated in Figure 10. The fiber volume fraction (V_f_) of the composites was calculated to be close to 60 ± 2%. Based on that, the incorporation of the SHA did not affect the thickness and architecture of the composite. Five samples per material type were tested under Mode I loading conditions (prior to and after the healing activation), while one sample left for cross-sectional OM examinations. During the manufacturing process, two polytetrafluoroethylene sheets of 13 μm thickness were positioned into the mid-thickness area to create an artificial pre-crack.

The applied curing profile was the following: 100 °C of applied temperature, 4 bar of applied pressure for 5 h. After curing, all CFRPs were ultrasonically scanned by using a physical acoustics corporation (UT C-scan system) with a 5 MHz transducer. Figure 9c illustrates C-scan plots for all materials manufactured. Finally, the OM examination of their morphology was performed by using the SINOWON IMS-300 microscope (Dongguan, China).

### 3.5. Mode I Testing

The tests were conducted according to the AITM 1.0005 standard of Airbus [30]. The information that is related to the tests is provided in a recent paper by the authors [13].

### 3.6. Healing Procedure, Healing Efficiency (H.E.) Calculations and Differential Scanning Calorimetry Test (DSC)

After the mechanical testing, the purposefully delaminated modified specimens were subjected to a simple healing cycle that consisted of heating at 130 °C for 30 min under uniform through-the-thickness compression of 1 kN (0.16 bar), using a conventional heat press machine. Heating of the system at elevated temperature is required in order for the retro DA reaction to take place. After healing, the samples were left to cool down for several hours; during the cooling step, the DA reaction took place. After the healing process, the samples were tested again under the same identical loading conditions. The H.E. achieved was calculated from Equation (1).
(1)$$H.E.=\frac{a_{\mathrm{healed}}}{a_{\mathrm{modified}}}\cdot 100\;\left(\%\right)$$
where a is the property under examination (i.e., P_max_ or G_IC_), a_healed_ refers to the value of the property after healing and a_modified_ refers to the property before any healing process. Based on previous publications [4,16] CFRPs’ structural integrity and mechanical properties are not affected by the application of the precise healing cycle.

In Figure 11, differential scanning calorimetry (DSC) curve is provided in order for the R-DA temperature of the utilized SHA to be identified. According to this test, it was shown that the R-DA temperature was calculated to be close to 110 °C. The DSC test was performed by using the Perkin-Elmer DSC 8500 calorimeter and the SHA sample was heated from RT to 250 °C at a rate of 5 °C/min.

### 3.7. Three-Point Bending Testing

The 3PB tests were performed according to ASTM D7264M-07 [31] in an Instron test frame. Test information have already been reported in [22].

## 4. Conclusions

The present study deals with the fabrication of new healable CFRP structures by using the MEP technique. More precisely, DA-based polymer (cross-linked BMI) was successfully synthesized and further utilized to modify pre-preg plies in a pure and nano-modified form (by incorporating 1 wt% GNPs). Based on that, two types of modified CFRPs were manufactured by incorporating one surface-modified pre-preg ply (locally) into the CFRPs’ mid-thickness area. In parallel, reference CFRPs were also manufactured (without SHA). After the manufacturing process, all CFRPs were tested under Mode I loading conditions and further assessed. Based on the experimental results, the modified CFRPs exhibited considerably increased Mode I interlaminar fracture toughness properties with BMI- and GNP-modified ones to present the best toughening performance (P_max_ value increased by almost 130% value while G_IC_ value by almost 285%). R-curves revealed the capacity of SHA itself and GNPs to enhance the fracture toughness properties, as both modified CFRPs exhibited extended DPZ against the reference one. The improved behavior is mainly in reference to the bridging effect during testing due to the presence of the SHA into the mid-thickness area of the DCB samples.

After fracture, modified samples passed through the healing process in order to be tested again and to further the healing capability to be investigated. Based on that, Mode I tests were repeated under the same conditions with the aim of calculating the H.E. values of the fracture toughness I properties. According to calculations, it was shown that all modified samples presented healing behavior with BMI- and GNP-modified samples to present the best healing performance. More precisely, P_max_ and G_IC_ values were restored at the amount of 62% and 45%, respectively. This behavior is mainly attributed to the thermally conductive phase (GNPs) that was integrated into SHA material. Furthermore, 3PB tests were conducted to investigate the effect of SHA incorporation on the in-plane mechanical properties of the CFRPs. According to the tests, in was shown that the in-plane mechanical properties were not significantly affected. Finally, OM observations revealed the extended fiber bridging effect at the fractured surfaces that occurred between the crack flanks during the tests as several pulled-out fibers were detected.

## Figures and Tables

**Figure 1 ijms-23-03663-f001:**
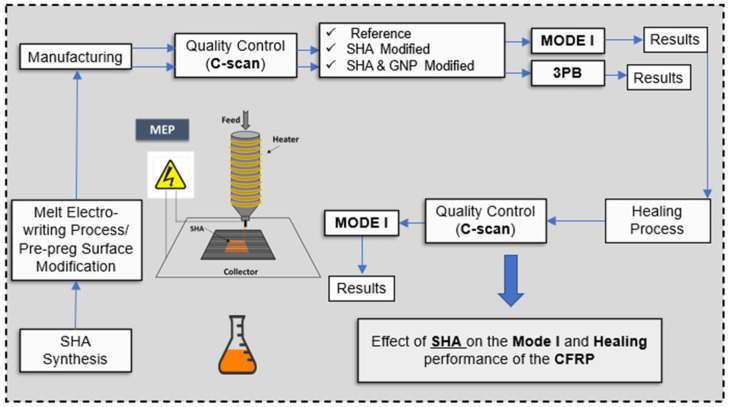
Schematic presentation of the scheduled characterization campaign conducted in the present investigation.

**Figure 2 ijms-23-03663-f002:**
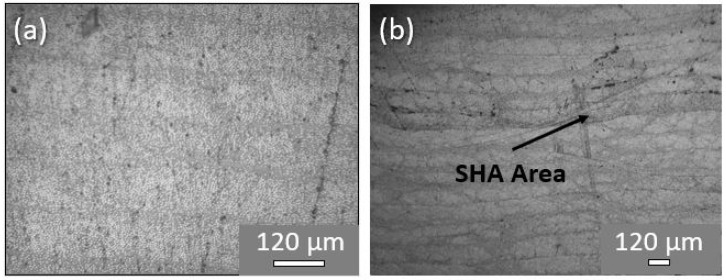
Representative optical microscopy images from cross-sections of (**a**) a reference and (**b**) a representative modified CFRP.

**Figure 3 ijms-23-03663-f003:**
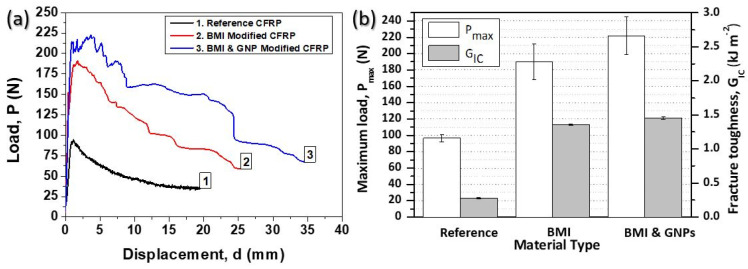
(**a**) Representative load (P) vs. displacement (d) curves for the reference and the two modified CFRPs, (**b**) synopsis of the Mode I testing results regarding the peak load (P_max_) and the fracture toughness energy I (G_IC_) values.

**Figure 4 ijms-23-03663-f004:**
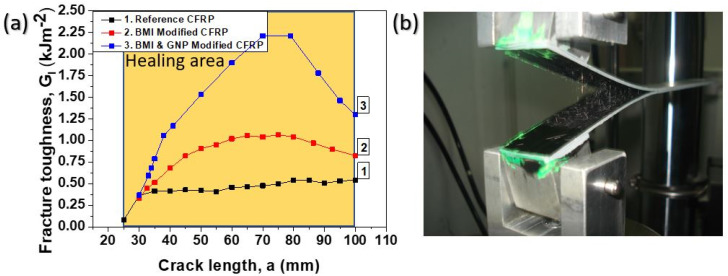
(**a**) Typical crack opening resistance curves (R-curves) under mode I loading conditions for the reference and modified CFRPs with pure BMI and BMI and GNPs. (**b**) Snapshot of a mode I test specimen (DCB) during testing.

**Figure 5 ijms-23-03663-f005:**
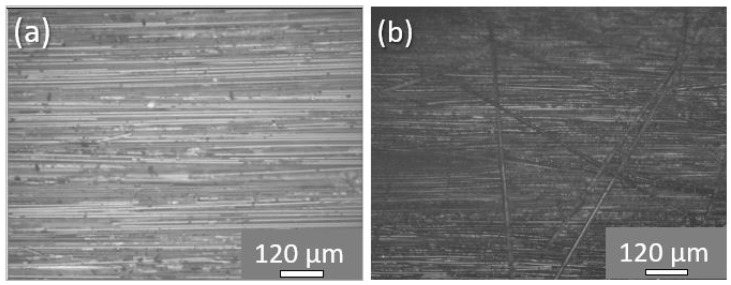
Representative optical microscopy (OM) images from the fractured surface of (**a**) a reference and (**b**) a representative modified CFRP, respectively.

**Figure 6 ijms-23-03663-f006:**
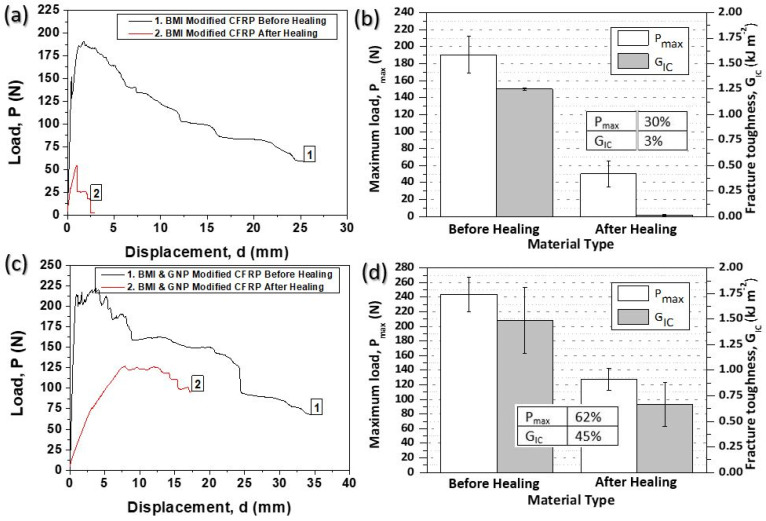
(**a**,**c**) Representative load (P) vs. displacement (d) curves for the two modified CFRPs, respectively before and after the activation of the healing process, (**b**,**d**) synopsis of Mode I testing results regarding the peak load (P_max_) and fracture toughness energy I (G_IC_) values together with healing efficiencies (H.E.) for modified CFRPs with BMI and BMI and GNPs, respectively.

**Figure 7 ijms-23-03663-f007:**
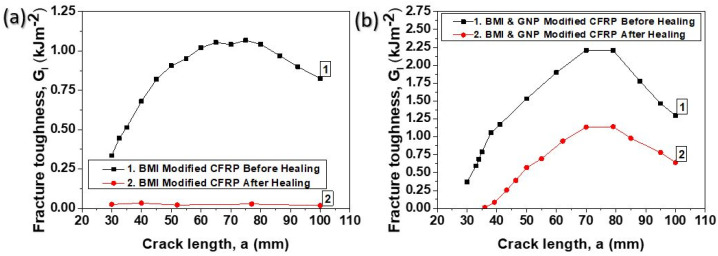
Typical crack opening resistance curves (R-curves) under mode I loading conditions for the modified CFRPs with (**a**) pure BMI and (**b**) BMI and GNPs, before and after the activation of the healing cycle.

**Figure 8 ijms-23-03663-f008:**
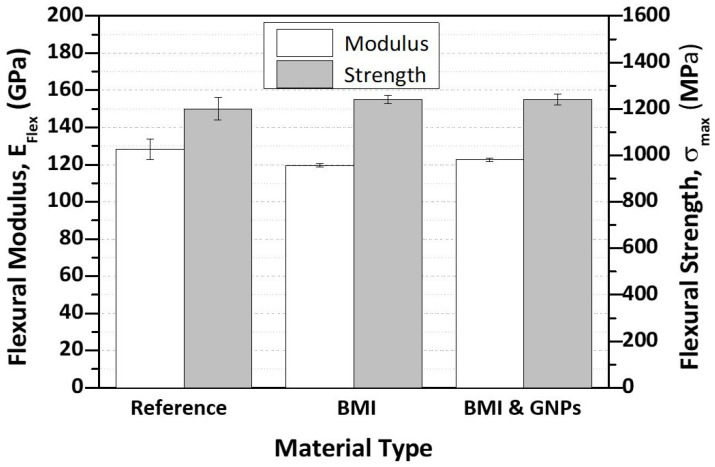
Bar chart diagram providing the flexural properties of the reference and of the two modified CFRPs.

**Figure 9 ijms-23-03663-f009:**
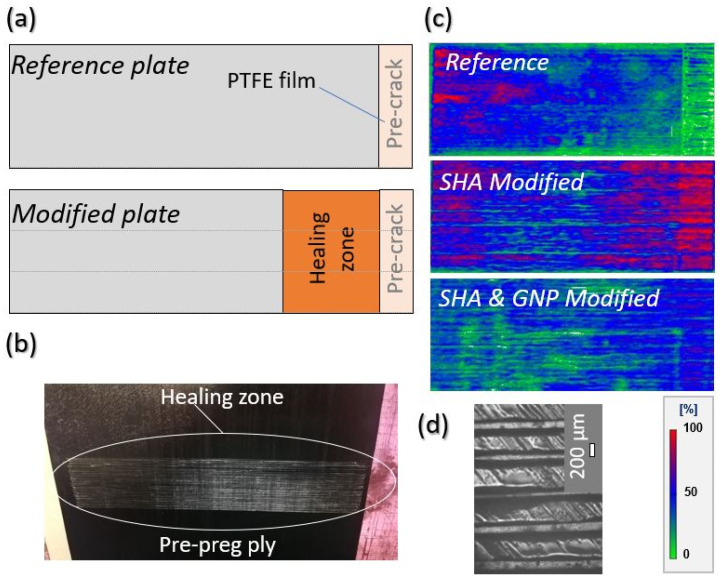
(**a**) Design of the reference and of the modified plates. (**b**) Depiction of the SHA surface-modified pre-preg ply by the melt electro-writing process (MEP) technique. (**c**) C-scan inspection images of all material sets (reference and modified ones). (**d**) Optical microscopy image, providing the fibrous mesh morphology employed for this study.

**Figure 10 ijms-23-03663-f010:**
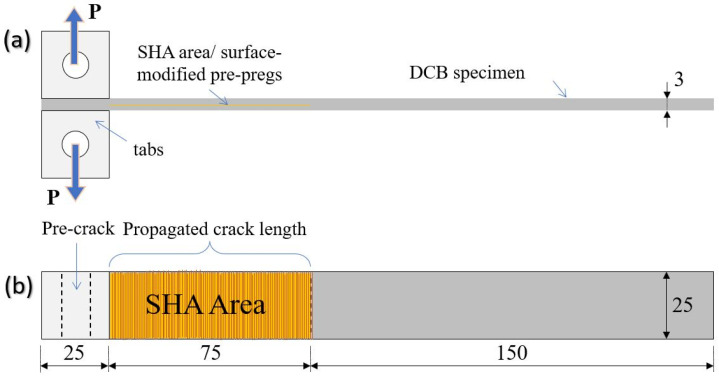
Illustration of a modified double cantilever beam (DCB) test configuration: (**a**) side view; (**b**) top view. Dimensions in millimeter (mm).

**Figure 11 ijms-23-03663-f011:**
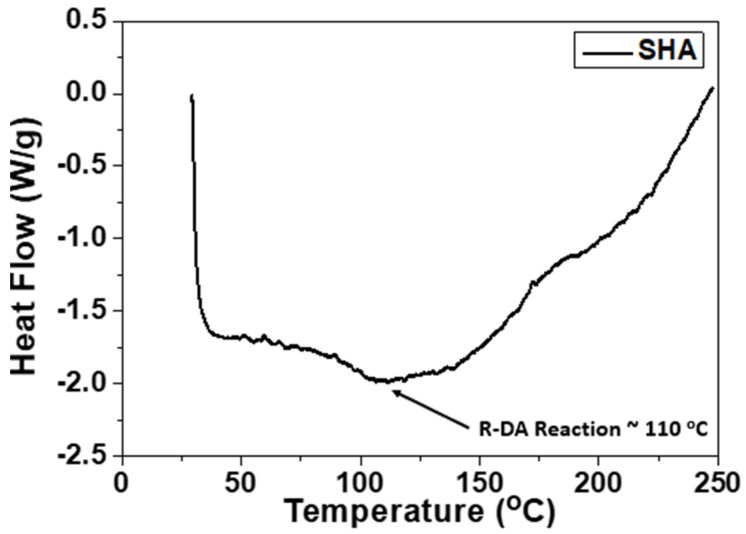
Differential scanning calorimetry (DSC) curve for the Diels–Alder (DA) mechanism-based SHA, utilized for the needs of the current study.

## Data Availability

Not applicable.

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
