# Peer review of "Toughening and Healing of CFRPs by Diels–Alder-Based Nano-Modified Resin through Melt Electro-Writing Process Technique"

_ijms, 2022, doi:10.3390/ijms23073663_

Round 1
Reviewer 1 Report
In this manuscript, the authors developed a new approach in which trifuran self-healing agent and graphene nanoplatelets were introduced into unidirectional carbon fiber reinforced plastics. The modified carbon fiber reinforced plastics revealed significant enhancements of the interlaminar fracture toughness properties. The overall data provides a comprehensive elaboration for the modification methodology. I believe that readers of International Journal of Molecular Sciences may be interested in this work. However, a major revision is needed prior to publication:
- For introducing the background of self-healing agents, the authors almost exclusively cited their own papers. Additional references of extrinsic and intrinsic self-healing materials from other research groups should be cited.
- The SHA reactant name " trifurane" should be corrected to " trifuran".
- The chemical structure of trifuran should be identified by NMR and FTIR.
- Why choose the adding amount of 1wt% graphene nanoplatelets? The author should explain the reason.
- The retro DA reaction should be identified by DSC.
- Figure 2(b) shows the representative cross-section image of modified CFRPs. Which sample does this image belong to (with or without GNPs)? The authors should present the cross-section images of both the modified CFRPs (with and without GNPs) and compare the differences in detail (between reference, SHA, and SHA/GNPs CFRPs).
- Similar to the previous question, Figure 5 should present the fractured surface images of the reference, SHA, and SHA/GNPs CFRPs, and a detailed discussion of these results is also needed.
- The sample name "BMI & MWCNTs" in Figure 8 should be corrected to "BMI & GNPs".

Author Response
In this manuscript, the authors developed a new approach in which trifuran self-healing agent and graphene nanoplatelets were introduced into unidirectional carbon fiber reinforced plastics. The modified carbon fiber reinforced plastics revealed significant enhancements of the interlaminar fracture toughness properties. The overall data provides a comprehensive elaboration for the modification methodology. I believe that readers of International Journal of Molecular Sciences may be interested in this work. However, a major revision is needed prior to publication:
Answer:
The authors would like to thank a lot reviewer #1 for the valuable comments.
- For introducing the background of self-healing agents, the authors almost exclusively cited their own papers. Additional references of extrinsic and intrinsic self-healing materials from other research groups should be cited (Ref. 8, 9, 23,29).
Answer: Four more references from other groups (related to extrinsic and intrinsic self-healing type) has been incorporated to the reference list.
- The SHA reactant name " trifurane" should be corrected to " trifuran".
Answer: The SHA reactant name has been corrected.
- The chemical structure of trifuran should be identified by NMR and FTIR.
Answer: The synthesized trifuran by the authors has been investigated under NMR and FTIR tests and it was proved that the resulted material is identical to relative literature which taken into consideration for fabrication. Relative information is provided into the manuscript, section 3.2.
- Why choose the adding amount of 1wt% graphene nanoplatelets? The author should explain the reason.
Answer: In all the previous, where cases we used nano-modified resin materials for increasing toughness the %wt used for having the highest/optimum results was 1%. This result that drove us to use 1% wt nanoparticles.
- The retro DA reaction should be identified by DSC.
Answer: DSC test has been performed and incorporated into the text as Figure 11, in order the Retro-DA reaction to be identified. In addition, the specifications of the test are fully provided into the section 3.6.
- Figure 2(b) shows the representative cross-section image of modified CFRPs. Which sample does this image belong to (with or without GNPs)? The authors should present the cross-section images of both the modified CFRPs (with and without GNPs) and compare the differences in detail (between reference, SHA, and SHA/GNPs CFRPs).
Answer: As it is already mentioned within the manuscript, there is no difference between the cross-sectional optical microscopy images of the two modified CFRPs as optical microscopy macroscopically examines the entire composite. The incorporation of 1 wt% GNPs into the SHA does not make a difference. This is the reason why the authors have chosen to show a representative image.
- Similar to the previous question, Figure 5 should present the fractured surface images of the reference, SHA, and SHA/GNPs CFRPs, and a detailed discussion of these results is also needed.
Answer: Based on previous answer, the fractured surface images arose from optical microscopy for both modified CFRPs provide identical images. This is the reason why the authors have also chosen to present a representative one.
- The sample name "BMI & MWCNTs" in Figure 8 should be corrected to "BMI & GNPs".
Answer: Figure 8 has been replaced by the corrected one.
Reviewer 2 Report
This manuscript well described the enhancement of toughening and self-healing of CFRPs using SHA (with or without GNPs). There are a few comments for the minor revision.
- In Abstract, "Mode I" should be specified.
- In Figures 3 and 4, it will be better to additionally describe physical meanings why the SHA significantly enhanced toughening properties and crack opening resistance properties.
- In Figure 5, the scales (120um and 40um) of two images were different each other. Also, how can we distinguish fracture toughness properties from OM images in Fig. 5?
- Compared to other results, Flexural modulus data were almost similar for three samples in Fig. 8. Is there any major reason?
Author Response
This manuscript well described the enhancement of toughening and self-healing of CFRPs using SHA (with or without GNPs). There are a few comments for the minor revision.
Answer:
The authors would like to thank a lot reviewer #2 for the valuable comments.
- In Abstract, "Mode I" should be specified.
Answer: The term “Mode I” has been specified for more accuracy.
- In Figures 3 and 4, it will be better to additionally describe physical meanings why the SHA significantly enhanced toughening properties and crack opening resistance properties.
Answer: The reason why the SHA significantly caused enhancement of the Mode I interlaminar fracture toughness properties of the modified composites is mainly attributed to the thermoplastic nature of it that caused the bridging effect. Modified CFRPs exhibited more ductile fracture behaviour against reference one. Based on that, more energy is required during testing due to the plastic deformation of the SHA itself. Relative information has been incorporated into 2.3. section.
- In Figure 5, the scales (120um and 40um) of two images were different each other. Also, how can we distinguish fracture toughness properties from OM images in Fig. 5?
Answer: Figure 5b has been replaced with a new one having the same scale as Figure 5a. We cannot distinguish the fracture toughness properties from an OM image. The only thing that we can distinguish is the bridging effect that took place during Mode I testing and means that the Mode I fracture toughness properties of the modified CFRPs have significantly been affected/improved. The Figure 5b is a “proof” of the bridging phenomenon that occurred during testing.
- Compared to other results, Flexural modulus data were almost similar for three samples in Fig. 8. Is there any major reason?
Answer: The Flexural properties for all material sets were expected to be similar as the SHA material was incorporated locally (only into the mid-thickness area/neutral zone) having a form of a very thin structure “fibrous interlayer”. Based on these, the Flexural properties were calculated to be very close to the properties of the Reference material. Relative explanations are provided in Section 2.5.

Round 2
Reviewer 1 Report
The authors have addressed and clarified the related issues. I recommend this manuscript for publication in IJMS.